# Make Your LLM Fully Utilize the Context

**Shengnan An**[*,◇,♣], **Zexiong Ma**[*,♡,♣], **Zeqi Lin**[†,♣],
**Nanning Zheng**[†,◇], **Jian-Guang Lou**[♣], **Weizhu Chen**[♣]

◇National Key Laboratory of Human-Machine Hybrid Augmented Intelligence,
National Engineering Research Center of Visual Information and Applications,
Institute of Artificial Intelligence and Robotics, Xi'an Jiaotong University
♣Microsoft ♡Peking University

◇{an1006634493@stu,nnzheng@mail}.xjtu.edu.cn,
♡mazexiong@stu.pku.edu.cn, ♣{Zeqi.Lin,jlou,wzchen}@microsoft.com

## Abstract

While many contemporary large language models (LLMs) can process lengthy input, they still struggle to fully utilize information within the long context, known as the *lost-in-the-middle* challenge. We hypothesize that it stems from insufficient explicit supervision during the long-context training, which fails to emphasize that any position in a long context can hold crucial information. Based on this intuition, our study presents INformation-INtensive (IN2) training, a purely data-driven solution to overcome lost-in-the-middle. Specifically, IN2 training leverages a synthesized long-context question-answer dataset, where the answer requires (1) **fine-grained information awareness** on a short segment (∼128 tokens) within a synthesized long context (4K−32K tokens), and (2) the **integration and reasoning** of information from two or more short segments. Through applying this information-intensive training on Mistral-7B, we present **FILM-7B** (**FIL**l-in-the-**M**iddle). To thoroughly assess the ability of FILM-7B for utilizing long contexts, we design three probing tasks that encompass various context styles (document, code, and structured-data context) and information retrieval patterns (forward, backward, and bi-directional retrieval). The probing results demonstrate that FILM-7B can robustly retrieve information from different positions in its 32K context window. Beyond these probing tasks, FILM-7B significantly improves the performance on real-world long-context tasks (e.g., 23.5→26.9 F1 score on NarrativeQA), while maintaining a comparable performance on short-context tasks (e.g., 59.3→59.2 accuracy on MMLU).

## 1 Introduction

> *To a great mind, nothing is little.*
>
> —*Arthur Conan Doyle*

Long-context large language models (LLMs) have recently received significant attention within the open-source community (Jiang et al., 2023; Du et al., 2022; Li et al., 2023a; Shi et al., 2023; Team et al., 2023; Team, 2023; Chen et al., 2023a; Song et al., 2023; Liu et al., 2023; Peng et al., 2023b; Chen et al., 2023b; Xiong et al., 2023; Tworkowski et al., 2024; AI et al., 2024; Ding et al., 2024; Mohtashami & Jaggi, 2024; Fu et al., 2024; Cai et al., 2024; Bai et al., 2024; Lv et al., 2024). The training context windows of many contemporary LLMs have been expanded to tens of thousands of

---

[*] Work done during the internship at Microsoft.
[†] Corresponding authors.

38th Conference on Neural Information Processing Systems (NeurIPS 2024).

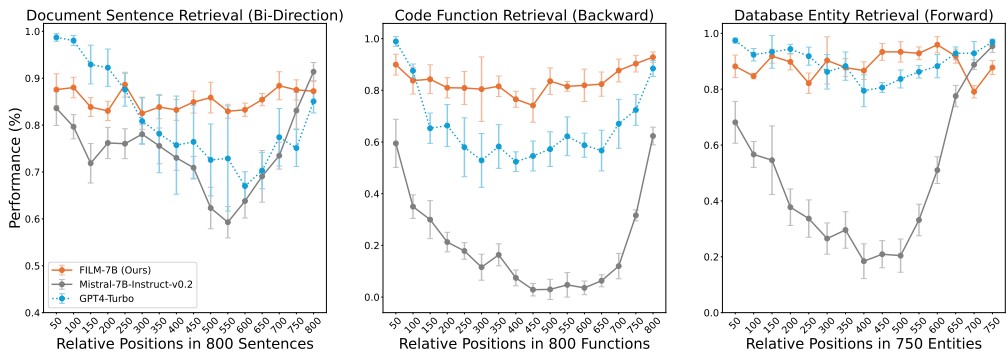

Figure 1: Performance of FILM-7B, Mistral-7B-Instruct-v0.2, and GPT4-Turbo on our three probing tasks. FILM-7B significantly overcomes the problem of information loss in the middle of the context.

tokens, thereby enabling these models to process extensive context as input. This extended training context window can enhance many real-world downstream tasks such as long-context question answering (Kočiský et al., 2018; Dasigi et al., 2021; Bai et al., 2023) and summarization (Fabbri et al., 2019; Huang et al., 2021; Zhong et al., 2021).

However, recent studies have revealed that these long-context LLMs struggle to effectively and robustly utilize all the information provided in the context, known as the *lost-in-the-middle* challenge (Liu et al., 2024b; Xu et al., 2023). It implies that while the LLM can comprehend the information at the beginning and end of the long context, it often overlooks the information in the middle. This challenge could significantly hinder the development of long-context LLMs, as they even often fail to pass simple probing tasks such as Needle-in-the-Haystack and passkey retrieval (Mohtashami & Jaggi, 2024). Consequently, a pressing research question arises: *how can we make long-context LLMs fully utilize the information in the long context?*

We hypothesize that the root cause of lost-in-the-middle stems from the unintentional bias hidden in the general training data. In auto-regressive pre-training, the loss on predicting the next token is more likely to be influenced by a few nearby pre-tokens rather than long-distance tokens (Sharan et al., 2018; Sun et al., 2021). For supervised fine-tuning and alignment, the system message, which strongly influences the generation of the response, is typically presented at the beginning of the context (Touvron et al., 2023; Cai et al., 2024). As a result, the general training process may inadvertently introduce a position bias, suggesting that important information is always located at the beginning and end of the context.

Based on this hypothesis, our work introduces **INformation-INtensive (IN2) training** to explicitly teach the model that **the crucial information can be intensively present throughout the context**, not just at the beginning and end. IN2 training is a purely data-driven solution that utilizes a synthesized long-context question-answer dataset. The long context (ranging from 4K to 32K tokens) is concatenated from many short segments (∼128 tokens), and the question-answer (QA) pairs ask for the information contained in one or more segments which are *randomly* placed in the long context. Specifically, we generate two types of questions, requiring (1) **fine-grained information awareness** on exactly one short segment, and (2) the **integration and reasoning of information** from two or more segments. These QA pairs are generated by prompting GPT-4-Turbo (OpenAI, 2023b) with the designed instructions and the raw segments.

By applying this information-intensive training on Mistral-7B (Jiang et al., 2023), we present **FILM-7B** (**FIL**l-in-the-**M**iddle). To thoroughly assess the long-context information awareness of FILM-7B, we design three probing tasks encompassing various context styles (document, code, and structured-data context) and information retrieval patterns (forward, backward, and bi-directional retrieval). The probing results (Figure 1) demonstrate that IN2 training significantly overcomes the lost-in-the-middle problem for the backbone model. Moreover, it can enhance the open-source model to achieve comparable or even more robust performance compared with proprietary LLMs such as GPT-4-Turbo.

Beyond these probing tasks, the performance of FILM-7B on real-world long-context tasks also exhibits significant improvements (e.g., 23.5→26.9 F1 score on NarrativeQA (Kočiský et al., 2018)). This demonstrates that the post-training on synthesized long-context data can be generalized to

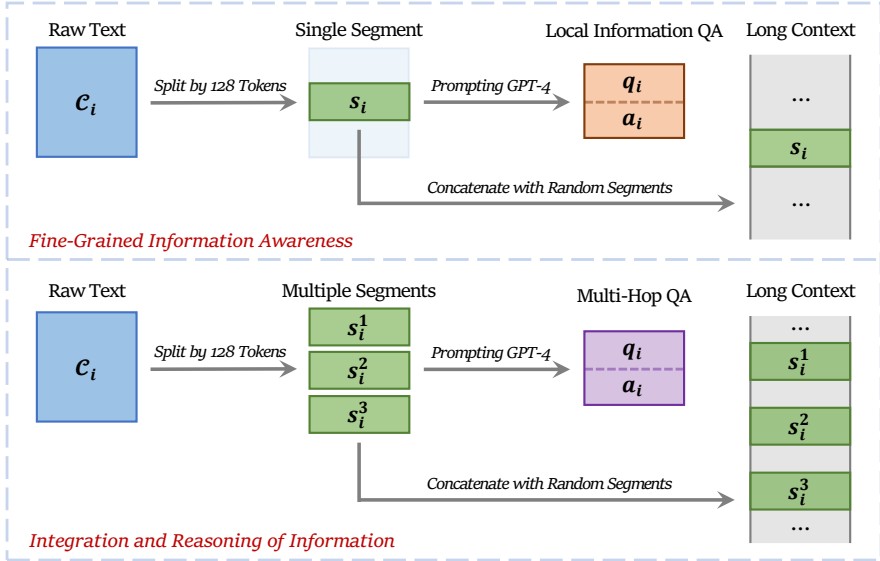

Figure 2: The data construction process for IN2 training, aimed at enhancing the fine-grained information awareness (upper), and the integration and reasoning of information (lower).

real-world scenarios. Moreover, FILM-7B maintains a comparable performance on short-context tasks compared with the vanilla backbone model (e.g., 59.3→59.2 accuracy on MMLU (Hendrycks et al., 2020)). This indicates that the short-context capability of FILM-7B is not compromised during training. Our further analysis explores how the sliding window strategy and the choice of RoPE base $\theta$ influence the performance of IN2 training.

## 2 Information-Intensive Training

This section introduces the construction of the dataset for IN2 training and the detailed training process of our model FILM-7B.

### 2.1 Training Data Construction

**Overview.** The IN2 training aims to explicitly teach the model that any position in a long context can contain crucial information. To achieve this goal, we construct a long-context question-answer training dataset $\mathbb{D} = \{\mathcal{L}_i, q_i, a_i\}$, where the answer $a_i$ to the question $q_i$ requires the information contained in some short segments that are randomly placed in the whole long context $\mathcal{L}_i$.

Figure 2 illustrates an overview of the data construction process. Specifically, the training data $\mathbb{D}$ is constructed based on a general natural language corpus $\mathbb{C}$. Given a raw text $\mathcal{C}_i \in \mathbb{C}$, we first generate a question-answer pair $(q_i, a_i)$ using a powerful LLM, then synthesize a long context $\mathcal{L}_i$ that includes the necessary information from $\mathcal{C}_i$ and other randomly sampled texts from $\mathbb{C}$. We generate two types of question-answer pairs that require (1) the awareness of fine-grained information in the long context, and (2) the integration and reasoning of information appearing at different positions in the long context. We take the `realnewslike` subset from the C4 corpus (Raffel et al., 2020) as $\mathbb{C}$, and take GPT-4-Turbo (OpenAI, 2023b) as the LLM to generate QA pairs.

**Fine-grained information awareness.** We consider a 128-token segment as the minimum information unit of the context[3]. Given a raw text $\mathcal{C}_i$, we first randomly extract a 128-token segment $s_i$ from it, then generate the $q_i$, $a_i$ and $\mathcal{L}_i$ accordingly,

$$(q_i, a_i) \sim \text{Prompting}(s_i, I_f; \text{LLM}), \quad \mathcal{L}_i = \oplus\{\text{Shuffle}(s_i, [r_j])\}, \tag{1}$$

---

[3]Appendix D contains the implementation and our considerations for this design choice.

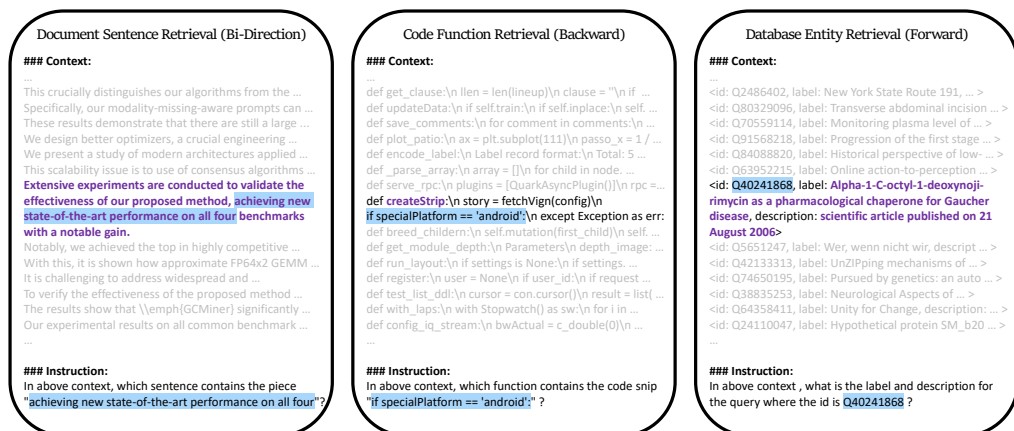

Figure 3: Three tasks in VAL Probing. The retrieval patterns are determined by the relative positions between the retrieval keywords and the **information to be retrieved**.

where $(q_i, a_i)$ is sampled by prompting the powerful LLM with the segment $s_i$ and the instruction $I_f$, $\oplus\{\cdot\}$ represents the concatenation of the contained segments, and $[r_j]$ are randomly sampled from 128-token segments in $\mathbb{C}$. Note that $I_f$ instructs the LLM to make the question-answer pair highly specific to the information provided in $s_i$.

**Integration and reasoning of information.** Beyond utilizing each single segment, we consider to generate question-answer pairs for information contained in two or more segments. Following the setting of the minimum information unit above, we split a full text $\mathcal{C}_i$ into a set of 128-token segments $[s_i]$, then generate the $q_i$, $a_i$ and $\mathcal{L}_i$ accordingly,

$$(q_i, a_i) \sim \text{Prompting}([s_i], I_r; \text{LLM}), \quad \mathcal{L}_i = \oplus\{\text{Shuffle}([s_i], [r_j])\}, \tag{2}$$

where $I_r$ instructs the LLM to generate a multi-hop question-answer pair that requires the information within at least two segments in $[s_i]$. All segments in $[s_i]$ and $[r_j]$ are jointly shuffled, so the required segments may appear far apart in the context.

**Context length balance and data mixture.** To prevent length bias during IN2 training, we ensure the length of the long context $\mathcal{L}_i$ is evenly distributed from 4K to 32K tokens. Such a length balance strategy can be implemented with restricted sampling on $[r_j]$, according to Equation 1 and 2. To alleviate catastrophic forgetting on short-context capabilities, we retain ∼10% question-answer pairs with the original texts $\mathcal{C}_i$ instead of converting them into a longer context, and add some general instruction-tuning data from the OpenOrca (Lian et al., 2023) dataset.

Overall, our dataset for IN2 training contains 1.1M long-context data for the fine-grained information awareness (∼63%), 300K long-context data for the integration and reasoning of information (∼17%), 150K short-context question-answer data (∼9%), and 200K general instruction-tuning data (∼11%). Appendix I contains the handcraft instructions for data generation. Appendix H illustrates some examples of our constructed long-context QA data. Appendix A describes the filtering strategy to avoid data contamination for evaluation.

## 2.2 Training Details

Using the training data constructed above, we further fine-tune the Mistral-7B-Instruct-v0.2[4] (Jiang et al., 2023) to get our FILM-7B (**FIL**l-in-the-**M**iddle). We perform IN2 training in the instruction-tuning paradigm: the long contexts and questions are used as instructions, and the loss on the answer parts are used to update the model. Appendix I contains the system template used for formatting the training data. For hyper-parameters, we set the global batch size as 128 and conduct one-epoch training with ∼14K training steps. We use the cosine learning rate decay with a 1e-6 maximum

---

[4]https://huggingface.co/mistralai/Mistral-7B-Instruct-v0.2.

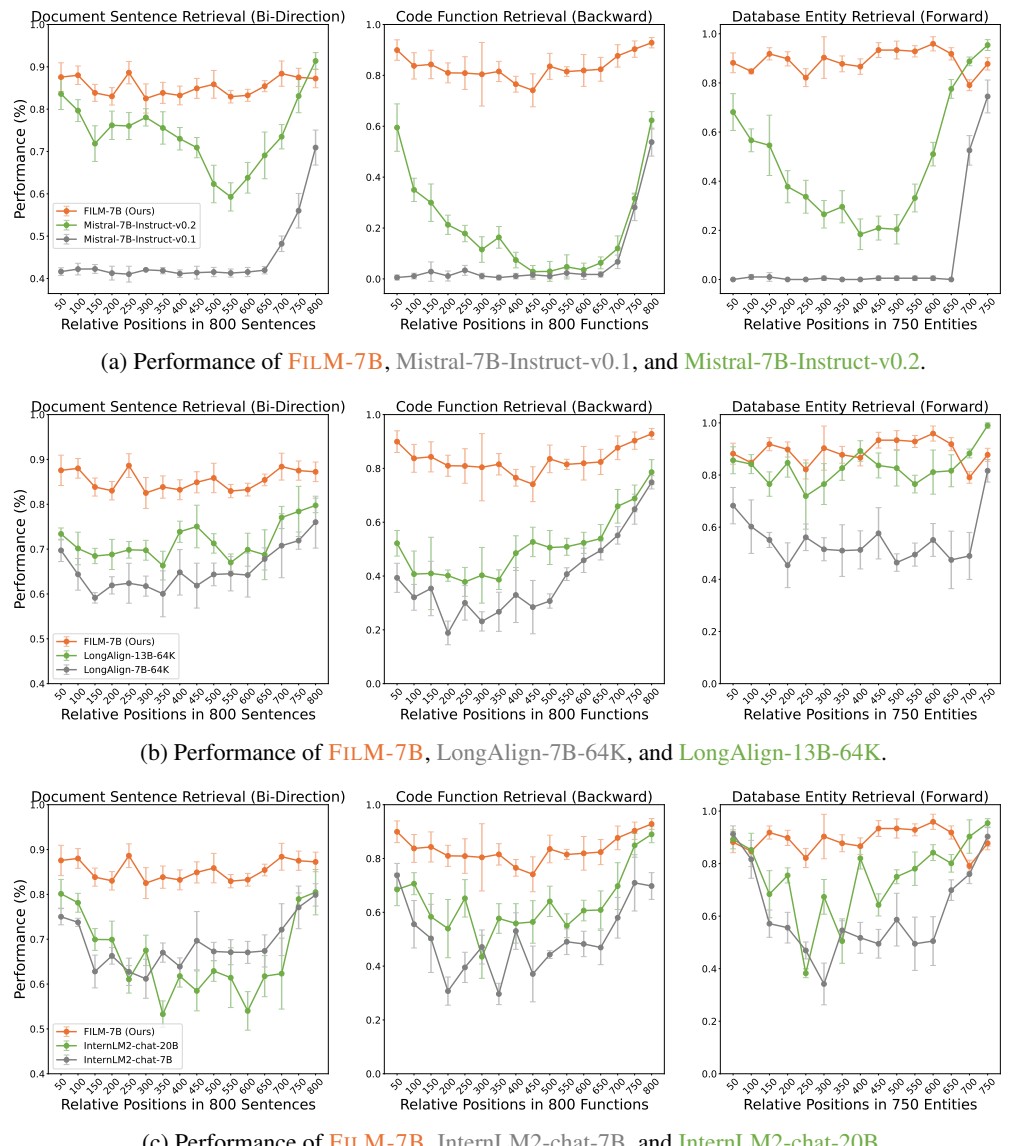

(a) Performance of FILM-7B, Mistral-7B-Instruct-v0.1, and Mistral-7B-Instruct-v0.2.

(b) Performance of FILM-7B, LongAlign-7B-64K, and LongAlign-13B-64K.

(c) Performance of FILM-7B, InternLM2-chat-7B, and InternLM2-chat-20B.

Figure 4: Performance of FILM-7B on VAL Probing and the comparisons with (a) Mistral, (b) LongAlign, and (c) InternLM2. The X-axis is the relative position in the context (∼32K tokens).

learning rate and 3% warm-up steps. The training process is conducted on 16 nodes of 8x80G A100 GPUs with the full sharding strategy and cpu offload strategy implemented by pytorch FSDP (Zhao et al., 2023). One entire training process (for a single FILM-7B model) consumes ∼300 GPU days.

# 3 Long-Context Probing

In this section, we first show the preliminary evaluation of FILM-7B on the Needle-in-the-Haystack and discuss about the inadequacies of this probing task. Subsequently, to comprehensively evaluate the long-context information awareness of FILM-7B, we introduce **VArious Long-context (VAL) Probing**. This includes three tasks that cover various context styles (document, code, and structured-data context) and information retrieval patterns (forward, backward, and bi-directional retrieval).

Table 1: Quantified performances of various models on VAL Probing.

| Model | Document | | Code | | Database | | All | |
|---|---|---|---|---|---|---|---|---|
| | Avg | Gap↓ | Avg | Gap↓ | Avg | Gap↓ | Avg | Gap↓ |
| Mistral-7B-Instruct-v0.1 (Jiang et al., 2023) | 44.8 | 29.9 | 6.8 | 53.2 | 8.8 | 74.5 | 20.1 | 52.5 |
| Mistral-7B-Instruct-v0.2 (Jiang et al., 2023) | 74.2 | 32.1 | 20.3 | 59.5 | 47.5 | 77.0 | 47.3 | 56.2 |
| LongAlign-7B-64K (Bai et al., 2024) | 65.3 | 16.9 | 39.3 | 56.0 | 55.0 | 36.2 | 53.2 | 36.4 |
| LongAlign-13B-64K (Bai et al., 2024) | 71.7 | 13.4 | 50.8 | 40.8 | 82.9 | 27.0 | 68.5 | 27.1 |
| InternLM2-chat-7B (Cai et al., 2024) | 68.8 | 18.7 | 50.2 | 44.1 | 61.2 | 57.1 | 60.1 | 40.0 |
| InternLM2-chat-20B (Cai et al., 2024) | 66.4 | 27.2 | 63.4 | 45.5 | 74.9 | 57.2 | 68.2 | 43.3 |
| GPT-4-Turbo (OpenAI, 2023b) | 81.3 | 31.7 | 66.1 | 46.5 | **89.6** | 18.0 | 79.0 | 32.1 |
| FILM-7B (ours) | **85.4** | **6.1** | **83.3** | **18.7** | 89.0 | **16.8** | **85.9** | **13.9** |

## 3.1 Near-Perfect Performance on Needle-in-the-Haystack: Are We There Yet?

The Needle-in-the-Haystack (Ivgi et al., 2023; Liu et al., 2024b) is widely used to assess how robustly a model utilizes information positioned in the long context. It reveals that even some powerful proprietary LLMs, such as GPT-4 and Claude 2.1 (Anthropic, 2023), struggle to fully exploit the information within the long context.

We use the Needle-in-the-Haystack task[5] to preliminarily evaluate the long-context capability of FILM-7B. Appendix B demonstrates that FILM-7B has achieved near-perfect performance on this task. This result is not surprising as recent open-source LLMs, such as LongAlign (Bai et al., 2024) and InternLM2 (Cai et al., 2024), have also shown near-perfect performance on this task.

However, the near-perfect performance on Needle-in-the-Haystack may overestimate the long-context capabilities of LLMs (Lei et al., 2024; Hsieh et al., 2024). Specifically, we have the following two concerns:

- Needle-in-the-Haystack employs a document-style context, which LLMs could be quite familiar with due to the pre-training on natural language corpora.
- The **forward retrieval** pattern in Needle-in-the-Haystack may simplify the difficulty of information seeking in the long context.

The "forward retrieval" means that the information being retrieved directly follows the retrieval keyword in a long context. For example, the default question used in Needle-in-the-Haystack is "What is the best thing to do in San Francisco?" and the answer is contained in "The best thing to do in San Francisco is eat a sandwich and sit in Dolores Park on a sunny day." The retrieved information "eat a sandwich and ..." just follows the retrieval keywords "best thing to do in San Francisco". According to the mechanism of induction head (Olsson et al., 2022), such a following-up copying is an easily learned pattern for LLMs, thus less challenging for evaluating long context utilization (just like the observation of "reversal curse" (Berglund et al., 2024)).

Given these considerations, we suggest that performances on Needle-in-the-Haystack may not adequately reflect the long-context capabilities of LLMs. Therefore, we propose VAL Probing for a more comprehensive evaluation involving various context styles and retrieval patterns.

## 3.2 VAL Probing

Our retrieval-based VAL Probing considers three context styles (document, code, and structured-data context) and three retrieval patterns (forward, backward, and bi-directional retrieval). Each context in VAL Probing contains ~32K tokens, and each task contains ~3K examples. Figure 3 briefly illustrates the contexts and retrieval instructions in VAL Probing.

**Document Sentence Retrieval (Bi-Direction).** The contexts consist of numerous natural language sentences, and the instruction aims to retrieve a single sentence containing a given piece. The sentences are sampled from the abstracts of papers on arXiv[6]. This task follows the bi-directional

---

[5]https://github.com/gkamradt/LLMTest_NeedleInAHaystack.
[6]https://info.arxiv.org/help/api/basics.html.

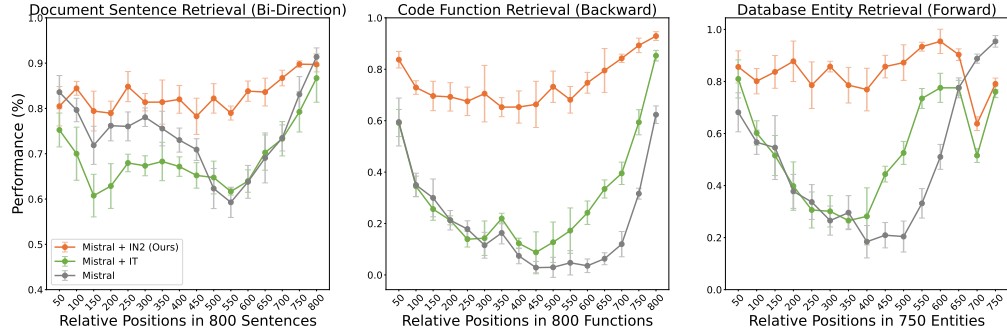

Figure 5: Compare the performance of IN2 training and general instruction tuning (IT). Both two training process takes the same number of training instances (20% of the full data size, 300K examples). Compare with Mistral + IN2 training, the gains from normal instruction tuning are marginal and unstable.

Table 2: Quantified comparison between IN2 training and normal instruction tuning.

| Model | Document | | Code | | Database | | All | |
|---|---|---|---|---|---|---|---|---|
| | Avg | Gap↓ | Avg | Gap↓ | Avg | Gap↓ | Avg | Gap↓ |
| Mistral-7B-Instruct-v0.2 (Jiang et al., 2023) | 74.2 | 32.1 | 20.3 | 59.5 | 47.5 | 77.0 | 47.3 | 56.2 |
| + Normal Instruction Tuning (Lian et al., 2023) | 69.0 | 25.9 | 30.2 | 76.5 | 53.4 | 54.4 | 50.9 | 52.3 |
| + Information-Intensive Training (ours) | **82.9** | **11.5** | **74.5** | **27.7** | **83.5** | **31.6** | **80.3** | **23.6** |

retrieval pattern, as the expected retrieval results contain words both before and after the given piece in the context. The evaluation metric is the word-level recall score.

**Code Function Retrieval (Backward).** The contexts consist of Python functions, and the instruction aims to retrieve the function name for a given line of code within the function definition. The raw code functions are sampled from the StarCoder (Li et al., 2023c) dataset[7]. We randomly select three lines of definitions for each function. This task follows the backward retrieval pattern, as the function name always precedes the definition. The evaluation metric is the exact-match accuracy.

**Database Entity Retrieval (Forward).** The contexts contain lists of structured entities, each with three fields: ID, label, and description. The query aims to retrieve the label and description for a given ID. The entities are sampled from Wikidata [8]. This task follows the forward retrieval pattern, as the label and description follow the ID. We take a relaxed exact-match accuracy as the metric: a 1 score is given if either the label or the description is exactly matched in the response, otherwise a 0 score.

## 4 Experiments and Analysis

We assess the long-context capability of FILM-7B on both probing tasks and real-world long-context tasks. Moreover, we investigate if the performance in short-context scenarios is affected.

### 4.1 Experimental Setup

**Models.** We mainly compare FILM-7B with long-context open-source models that have been trained with ≥32K context windows, including the Mistral (Jiang et al., 2023), LongChat (Li et al., 2023a), ChatGLM (Du et al., 2022), LongAlign (Bai et al., 2024), LongWanjuan (Lv et al., 2024), Yi (AI et al., 2024) and InternLM2 (Cai et al., 2024). We utilize the instruct/chat versions of these models as most of our evaluation tasks are under the zero-shot instruction-following paradigm. We also draw comparisons with popular proprietary LLMs such as GPT-3.5-Turbo (OpenAI, 2023a) and

---

[7]https://huggingface.co/datasets/bigcode/starcoderdata.
[8]https://www.wikidata.org/wiki/Wikidata:Data_access.

Table 3: Performances of various models on real-world long-context tasks. Results of models with * are reported in Bai et al. (2023) and Lv et al. (2024).

| Model | NarrativeQA | Qasper | MultiFQA | HotpotQA | 2WikiMQA | MuSiQue | GovReport | QMSum | MultiNews | Avg |
|---|---|---|---|---|---|---|---|---|---|---|
| Close-Source | | | | | | | | | | |
| GPT-4-Turbo (OpenAI, 2023b) | 33.0 | 50.7 | 52.7 | 68.5 | 64.3 | 49.1 | 33.9 | 25.4 | 24.9 | 44.7 |
| GPT-3.5-Turbo* (OpenAI, 2023a) | 23.6 | 43.3 | 52.3 | 51.6 | 37.7 | 26.9 | 29.5 | 23.4 | 26.7 | 35.0 |
| Open-Source | | | | | | | | | | |
| LongChat-v1.5-7B-32K* (Li et al., 2023a) | 16.9 | 27.7 | 41.4 | 31.5 | 20.6 | 9.7 | 30.8 | 22.7 | 26.4 | 25.3 |
| ChatGLM2-6B-32K* (Du et al., 2022) | 21.1 | 31.5 | 46.2 | 25.3 | 20.8 | 9.8 | 32.4 | 24.0 | 26.5 | 26.4 |
| LongAlign-7B-64K (Bai et al., 2024) | 18.7 | 33.8 | 49.1 | 28.6 | 23.4 | 12.5 | 30.6 | 23.7 | 27.5 | 27.5 |
| Mistral-7B-Instruct-v0.1 (Jiang et al., 2023) | 19.6 | 33.2 | 38.8 | 42.9 | 31.2 | 17.4 | 27.5 | 22.4 | 26.6 | 28.9 |
| Mistral-7B-Instruct-v0.2 (Jiang et al., 2023) | 23.5 | 33.8 | 45.9 | 42.4 | 24.3 | 20.8 | 33.3 | 24.8 | 26.8 | 30.6 |
| Yi-6B-200K* (AI et al., 2024) | 12.4 | 26.4 | 36.8 | 46.6 | 40.4 | 25.8 | 29.3 | 20.7 | 27.1 | 29.5 |
| ChatGLM3-6B-32K* (Du et al., 2022) | 9.2 | **43.1** | 50.9 | 55.3 | 43.7 | **38.9** | **36.0** | 24.7 | 27.4 | 36.6 |
| InternLM2-chat-7B (Cai et al., 2024) | 24.4 | 35.4 | 50.2 | 52.4 | **48.2** | 30.5 | 33.6 | **25.3** | **29.0** | 36.5 |
| InternLM2-7B-LongWanjuan* (Lv et al., 2024) | **29.9** | 39.6 | 50.2 | 53.7 | 42.3 | 32.1 | 33.0 | **25.5** | 27.8 | 37.1 |
| FILM-7B (ours) | 26.9 | 42.2 | **56.0** | **62.1** | 47.0 | **39.0** | 33.8 | 25.1 | 26.9 | **39.9** |

Table 4: Model performances on few-shot learning tasks.

| Model | TREC | TriviaQA | SAMSum | Average |
|---|---|---|---|---|
| GPT4-Turbo (OpenAI, 2023b) | 77.0 | 91.7 | 39.7 | 69.5 |
| Mistral-7B-Instruct-v0.2 (Jiang et al., 2023) | 71.0 | 84.5 | 35.8 | 63.8 |
| FILM-7B (ours) | **76.0** | **90.0** | **39.5** | **68.5** |

GPT-4-Turbo (OpenAI, 2023b). All models and tasks employ greedy decoding. For probing tasks, we primarily compare FILM-7B with LongAlign and InternLM2 series, as these models have shown near-perfect performances on Needle-in-the-Haystack.

**Real-world long-context tasks.** We take 9 tasks from the LongBench (Bai et al., 2023) collection to evaluate the long-context capability on real-world scenarios. These tasks encompass long-document question answering (NarrativeQA (Kočiskỳ et al., 2018), Qasper (Dasigi et al., 2021) and Multi-FieldQA (MultiFQA) (Bai et al., 2023), multi-document multi-hop reasoning (HotpotQA (Yang et al., 2018), 2WikiMultihopQA (2WikiMQA) (Ho et al., 2020) and MuSiQue (Trivedi et al., 2022)), and long-context summarization (GovReport (Huang et al., 2021), QMSum (Zhong et al., 2021) and MultiNews (Fabbri et al., 2019)). We employ the middle truncation strategy in LongBench to limit the input within 32K tokens. We report ROUGE-L (Lin, 2004) for summarization tasks and F1 scores for other tasks. The evaluation metrics are computed using the official evaluation scripts [9].

Despite these QA and summarization tasks, we also conduct evaluations on few-shot learning tasks, in which the contexts could also be extremely lengthy if there are many "few-shot" examples presented in the context. We take three in-context learning tasks from LongBench, including TREC (Li & Roth, 2002) for few-shot classification, TriviaQA (Joshi et al., 2017) for few-shot QA, and SAMSum (Gliwa et al., 2019) for few-shot summarization.

**Short-context tasks.** We select 8 short-context tasks commonly used for evaluating the general capabilities of models. These include MMLU (Hendrycks et al., 2020), BoolQ (Clark et al., 2019), RACE-High (RACE-H) (Lai et al., 2017), CommonsenseQA (CSQA) (Talmor et al., 2019), ARC-Challenge (ARC-C) (Clark et al., 2018), HellaSwag (Zellers et al., 2019), GSM8K (Cobbe et al., 2021), and MATH (Hendrycks et al., 2021). We use 5-shot for MMLU, 8-shot for GSM8K, 4-shot for MATH, and 0-shot for other tasks. We utilize the `lm_eval` (Gao et al., 2024) for the evaluations on MMLU, BoolQ, RACE-H, ARC-C and HellaSwag, and use the evaluation scripts from An et al. (2024) for other tasks.

### 4.2 Main Results and Analysis

**FILM-7B significantly mitigates the lost-in-the-middle problem.** Figure 4a presents the probing results for both FILM-7B and the backbone model, Mistral-7B-Instruct-v0.2. In all three probing tasks within VAL Probing, the vanilla Mistral model experiences substantial information loss at the middle positions in the long contexts. In contrast, our FILM-7B model consistently exhibits robust

---

[9]https://github.com/THUDM/LongBench.

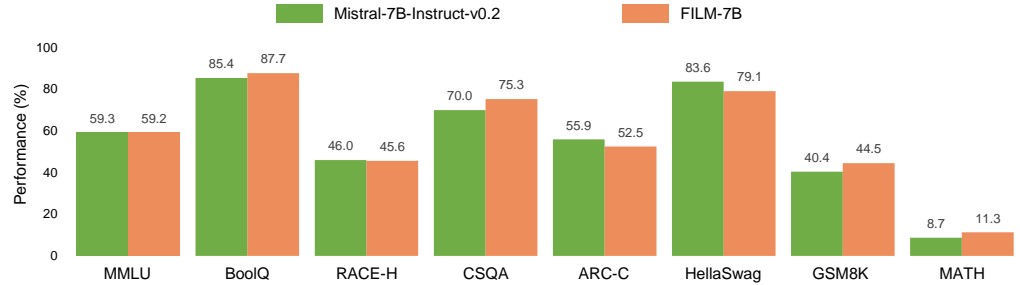

Figure 6: Performances of FILM-7B and the backbone model on short-context tasks.

performance across different positions within the whole context. This stark comparison illustrates that the lost-in-the-middle problem can be effectively addressed using our IN2 training.

**FILM-7B achieves performance comparable to, or even outperforming, that of GPT-4-Turbo.** Figure 1 illustrates the comparison between **FILM-7B** and GPT-4-Turbo on our probing tasks. Beyond a qualitative comparison between the performance curves of two models, we quantify the long-context performances on VAL Probing using two metrics:

- **Average score (Avg).** We compute the average performances across the entire context length, reflecting the overall long-context utilization.

- **Min-max gap (Gap).** We calculate the differences between the maximum and minimum performances in Figure 3. A smaller performance gap signifies greater robustness across different positions.

Table 1 presents the quantified performances on VAL Probing. It reveals that FILM-7B has comparable performance with GPT-4-Turbo on the database probing task, and exhibits better robustness in document and code probing tasks. These results indicate a great potential for the development of open-source long-context models to close the gap with proprietary models.

**IN2 training can effectively alleviate the lost-in-the-middle problem, while the normal instruction tuning cannot.** Considering that FILM-7B additionally uses instruction-tuning-style data for post-training, to further demonstrate the effectiveness of IN2 training, here we present more controlled experiments to compare IN2 training and normal instruction tuning under the same training data size. Specifically, for both IN2 training and normal instruction tuning, we take the same backbone model (i.e., Mistral-7B-Instruct-v0.2) and the same number of post-training instances (20% of our full data size, ~300K examples). The data for normal instruction tuning are randomly sampled from OpenOrca (Lian et al., 2023). Comparisons shown in Figure 5 shows the performance curves on VAL Probing after two training processes, and Table 2 contains the quantified results. These comparisons clearly demonstrate that IN2 training can effectively alleviate the lost-in-the-middle problem while the normal instruction tuning cannot.

**VAL Probing presents a more challenging test suite for long-context models.** Figure 4b and 4c show the probing results of LongAlign and InternLM2, two state-of-the-art long-context models. Despite their extended training context windows, these models still encounter the lost-in-the-middle problem. This is particularly noteworthy given their near-perfect performance on the Needle-in-the-Haystack task. This comparison suggests that VAL Probing provides a more challenging evaluation for long-context models.

In particular, the results on document and database tasks in VAL Probing demonstrate clear comparisons with Needle-in-the-Haystack. Compared to Needle-in-the-Haystack which uses forward retrieval on natural language context, the document task employs natural language context but with bi-directional retrieval, and the database task uses forward retrieval but with structured-data context. These comparisons highlight that both context styles and retrieval patterns significantly contribute to the hardness of the probing tasks.

**Training on synthesized long-context data effectively generalizes to real-world scenarios.** Table 3 and 4 contain the results on various real-world long-context tasks. It shows that FILM-7B also significantly improves the performance of the backbone model in real-world long-context scenarios. Moreover, it also achieves SOTA-level[10] performances on these tasks among ∼7B size open-source models. Notably, the long contexts used in IN2 training are all synthesized from short segments. These improvements suggest that the long-context capabilities learned from the synthesized data can be successfully applied to real-world tasks.

**FILM-7B maintains the performance on short-context tasks.** Figure 6 illustrates the performances of FILM-7B and the vanilla backbone model on short-context tasks. It reveals that the overall performances on short-context tasks are almost comparable with minor variances. These results confirm that FILM-7B does not compromise the short-context capabilities of the backbone model.

**Analysis on training strategies.** We are specifically interested in investigating the impact of the following two training strategies: applying the sliding window and adjusting the position encoding. Due to the page limitation, we provide these further ablations and analysis in Appendix C.

## 5 Related Work

**Long-context LLMs.** Recent research has significantly contributed to the exploration of training large models with extended context windows (Jiang et al., 2023; Du et al., 2022; Li et al., 2023a; Team et al., 2023; Team, 2023; Xiong et al., 2023; Song et al., 2023; Tworkowski et al., 2024; AI et al., 2024; Cai et al., 2024). There are primarily two directions in the development of long-context LLMs. (1) Data engineering, which emphasizes the construction of long-context data for training the LLMs. This includes data balancing (Fu et al., 2024), data order arrangement (Shi et al., 2023), instruction data collection (Bai et al., 2024), and data quality measurement (Lv et al., 2024). Our IN2 training can be categorized into this field. (2) Effective and efficient training, which investigates methods to optimize the training of a long-context model. This encompasses the design of position encoding (Chen et al., 2023a; Liu et al., 2023; Peng et al., 2023b; Ding et al., 2024), batching strategy (Bai et al., 2024), parameter-efficient training (Chen et al., 2023b), and the development of new model architectures (Peng et al., 2023a; Gu & Dao, 2023).

**Long-context evaluations.** Existing benchmarks for evaluating long-context models can be divided into two categories. (1) Real-world benchmarks that assess general long-context capabilities (e.g., long-context QA, summarization, and language modeling), such as NarrativeQA (Kočiskỳ et al., 2018), LongBench (Bai et al., 2023), ZeroSCROLLS (Shaham et al., 2023), L-Eval (An et al., 2023), Loogle (Li et al., 2023b), ∞Bench (Zhang et al., 2024), and a series of work on perplexity evaluation (Beltagy et al., 2020; Roy et al., 2021; Press et al., 2021; Chen et al., 2023a; Liu et al., 2023; Peng et al., 2023b; Chen et al., 2023b; Ding et al., 2024; Mohtashami & Jaggi, 2024). (2) Probing tasks that provide a more concise reflection of the long-context utilization across different context lengths and positions. These include Needle-in-the-Haystack, passkey retrieval (Mohtashami & Jaggi, 2024), synthesized document QA (Liu et al., 2024b), S3Eval (Lei et al., 2024), Discovery (Li et al., 2024), RULER (Hsieh et al., 2024), and the VAL Probing proposed in this study. Among these probing tasks, our VAL Probing is the first to explicitly incorporate a variety of retrieval patterns.

## 6 Conclusion

This work introduces IN2 training to overcome the lost-in-the-middle problem. By applying IN2 training on the open-source model, our FILM-7B exhibits significant improvements on probing tasks and real-world long-context tasks while does not compromise the short-context performance.

---

[10]The bold numbers in Table 3 are SOTA-level results among 7B open-source models. Specifically, for each task, we bold the highest result and the results within a margin to the highest one (0.5 for summarization tasks and 2.0 for others).

## Acknowledgments

We thank all the anonymous reviewers for their valuable comments. Shengnan An and Nanning Zheng were supported in part by NSFC under grant No. 62088102.

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

This is the Appendix of the paper: *Make Your LLM Fully Utilize the Context.*

## A Data Filtering Strategy

To avoid data contamination for the evaluation stage in Section 4, we apply a pre-filtering strategy during sampling the raw texts for constructing the dataset of IN2 training. Specifically, during sampling $C_i$ for generating data, if the sampled $C_i$ has a 10-gram overlap with any example in all of our evaluation data (including probing tasks, real-world tasks and short-context tasks), it will not be used for neither generating question-answer pairs nor serving as the random segments $[r_j]$.

## B Performance on Needle-in-the-Haystack

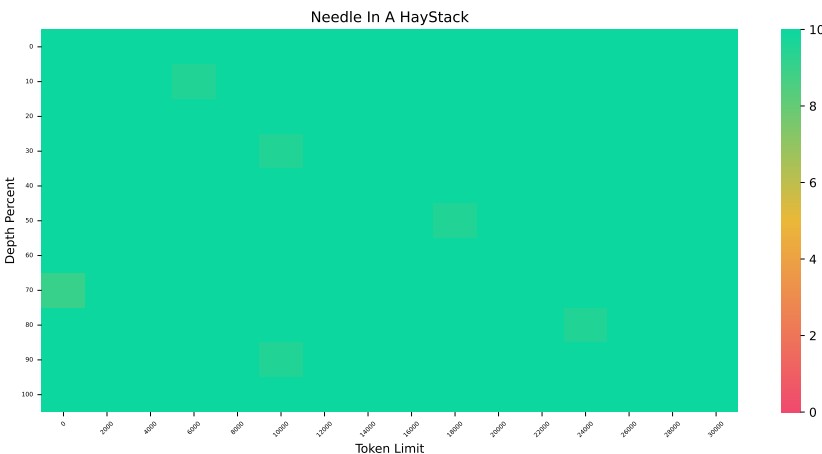

Figure 7: Performances of FILM-7B on Needle-in-the-Haystack.

Figure 7 shows the performance of FILM-7B on Needle-in-the-Haystack. It shows that FILM-7B has achieved near-perfect performance on Needle-in-the-Haystack within its 32K context window.

## C Training Strategy Analysis

Experimental results in Section 4.2 demonstrate the feasibility of IN2 training. We aim to explore further into enhancing the effectiveness and efficiency of IN2 training, particularly from the perspective of training strategies. We are specifically interested in investigating the impact of the following two training strategies: applying the sliding window and adjusting the position encoding. Considering the high cost of training, the following experiments use 20% of all training examples.

**Models using sliding windows cannot effectively capture the long distance information.** Our experiments involving Mistral models, as shown in Figure 4a, reveal that the performance of Mistral-7B-Instruct-v0.1 is awful when the information is positioned at a long distance. It's worth noting that Mistral-7B-Instruct-v0.1 employs the sliding window strategy while Mistral-7B-Instruct-v0.2 does not. Consequently, we are interested in determining whether our IN2 training can still alleviate the lost-in-the-middle problem under the sliding window strategy. We conduct the following two experiments with a 4K sliding window during training:

- **Apply the sliding window in both pre-training and IN2 training.** We take the Mistral-7B-Instruct-v0.1 as the backbone model and conduct IN2 training with the same window size (4K).
- **Apply the sliding window only during the IN2 training.** We take the Mistral-7B-Instruct-v0.2 as the backbone model and additionally apply a 4K sliding window during IN2 training.

Figure 8 illustrates the performances of models with sliding windows. It shows that in both two settings with sliding windows, the performances drop dramatically when the distance between the

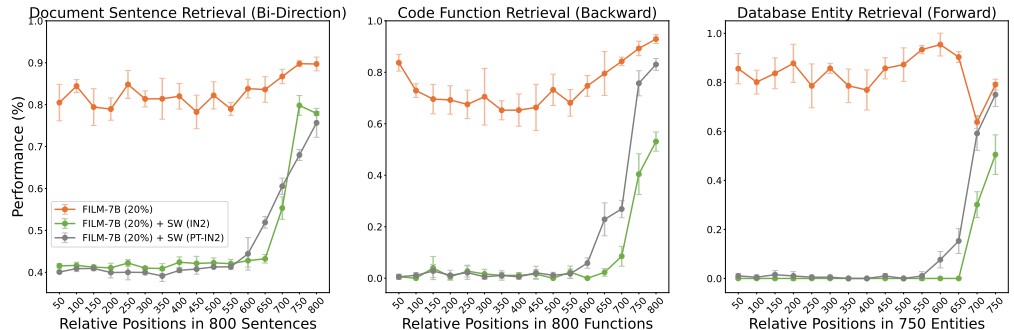

Figure 8: Performance of FILM-7B with a 4K sliding window (SW). PT-IN2: apply the sliding window in both pre-training and IN2 training. IN2: apply the sliding window only in IN2 training.

Table 5: Performance of FILM-7B with different RoPE base $\theta$ during IN2 training.

| Model | RoPE Base $\theta$ | Document | | Code | | Database | | All | |
|---|---|---|---|---|---|---|---|---|---|
| | | Avg | Gap↓ | Avg | Gap↓ | Avg | Gap↓ | Avg | Gap↓ |
| FILM-7B (20%) | $1.0 \times 10^6$ (default) | 82.9 | 11.5 | 74.5 | 27.7 | 83.5 | 31.6 | 80.3 | 23.6 |
| | $2.0 \times 10^6$ | 83.9 | 9.3 | 79.8 | 27.1 | 87.7 | **13.2** | 83.8 | 16.5 |
| | $1.0 \times 10^7$ | 83.7 | **7.6** | **81.7** | **18.4** | **89.4** | 16.8 | **84.9** | **14.3** |
| | $1.0 \times 10^8$ | **84.6** | 6.6 | 81.4 | 22.3 | 87.7 | **13.2** | 84.6 | **14.0** |

retrieval question and information is longer than the sliding window size. It reveals that the sliding window strategy greatly hurts the long-context capability of models.

**Training with higher information intensity requires a larger RoPE base $\theta$.** The training stage in Section 2 follows the RoPE settings configured for the backbone model. Previous studies on context extension suggest that training with an extended context length necessitates a larger RoPE base $\theta$ (Roziere et al., 2023; Xiong et al., 2023; Cai et al., 2024). In the case of our IN2 training, the context length remains unchanged, but the information density is significantly changed to be more uniform. There is a high-level similarity between context-extension training and our IN2 training: **both aim to enhance the model's capability to perceive information from a wider range of input positions** compared to the previous training stages. Consequently, we carry out experiments to investigate whether the experience from context-extension training could also benefit our IN2 training. Table 5 shows the results with increasing the RoPE base $\theta$ from $1.0 \times 10^6$ to $1.0 \times 10^8$. It shows that increasing the default RoPE base $\theta$ of the backbone model leads to better performances on VAL Probing. We suggest to use a 10 times of the default RoPE base $\theta$ to conduct IN2 training.

## D Implementation and Reasons for Segmentation

Algorithm 1 illustrates how we segment a raw text into ~128-token segments. We set the ~128-token segment as the minimum information unit due to the following consideration:

- If the segment contains too few tokens (e.g., 16 tokens or 32 tokens), it might not contain enough information for asking a meaningful question.

- If we set a large threshold for segmentation (e.g., 1024 tokens or 4096 tokens), most raw texts will just contain one segment[11], thus affecting the construction of QA pairs that require the integration and reasoning of information.

- Moreover, we do not just use a full raw text to generate a question with GPT-4, as we are concerned about whether GPT-4 will also more focus on the head and the tail of the text. It could result in a local bias for training: the answers are always placed on the boundaries between two segments containing consecutive information text.

---

[11]The raw texts in `realnewslike` have an average length of ~600 tokens with the Mistral tokenizer.

**Algorithm 1** Implementation of Raw Text Segmentation

---

**Given:**
  $\mathcal{C}_i$: The raw text;
  Tokjenizer($\cdot$): The tokenizer of the model;
  $l = 128$: The minimal length of each segment;
**Return:**
  $\mathcal{S}_i = [s_i^1, s_i^2, ...]$: A set of segments;
  1: $\mathcal{S}_i = [\,]$
  2: $\mathcal{P}_i = \text{Split}(\mathcal{C}_i, '\backslash n')$
  3: temp_list = [ ]
  4: temp_length = 0
  5: **for** $p_i^j \in \mathcal{P}_i$ **do**
  6:     length = Len(Tokenizer($p_i^j$)) temp_list.append($p_i^j$) temp_length += length
  7:     **if** temp_length >= $l$ **then**
  8:         temp_segment = '\n'.join(temp_list)
  9:         $\mathcal{S}_i$.append(temp_segment)
 10:         temp_list = [ ]
 11:         temp_length = 0
 12:     **else**
 13:         **continue**
 14:     **end if**
 15: **end for**
 16: **if** temp_list is not empty **then**
 17:     temp_list = [$\mathcal{S}_i$.pop()] + temp_list
 18:     temp_segment = '\n'.join(temp_list)
 19:     $\mathcal{S}_i$.append(temp_segment)
 20: **end if**
 21: **return** $\mathcal{S}_i$

---

Table 6: Performance of FILM-7B with different training data sizes for IN2 training.

| Model | Data Size | Document | | Code | | Database | | All | |
|---|---|---|---|---|---|---|---|---|---|
| | | Avg | Gap↓ | Avg | Gap↓ | Avg | Gap↓ | Avg | Gap↓ |
| | 100% | 85.4 | 6.1 | 83.3 | 18.7 | 89.0 | 16.8 | 85.9 | 13.9 |
| | 50% | 84.2 | 13.3 | 80.5 | 21.5 | 89.7 | 15.3 | 84.8 | 16.7 |
| FILM-7B | 20% | 82.9 | 11.5 | 74.5 | 27.7 | 83.5 | 31.6 | 80.3 | 23.6 |
| | 10% | 84.3 | 11.3 | 75.2 | 31.8 | 82.3 | 32.7 | 80.6 | 25.3 |
| | 1% | 76.2 | 18.8 | 63.3 | 48.0 | 70.9 | 36.7 | 70.1 | 34.5 |

Based on the above considerations, we use the $\sim$128-token segment. Due to the high cost on data generation, we do not conduct further ablations on this design choice.

## E   Data Scaling Trend

Table 6 shows the performance of FILM-7B with different training data sizes for IN2 training. Generally, with the data size increasing, the average performance increases and the performance variance in different positions decreases. Such trends are more significant on code and document probing tasks. Note that the training data for IN2 training are almost from natural language corpus. It indicates that increasing the training data size can better help the generalization of long-context capability on different context styles.

## F   Performance on RULER Benchmark

RULER (Hsieh et al., 2024) is a synthetic benchmark that evaluates the effective context length of the long-context LLMs. It revealed that while all existing long-context models with ≤7B sizes claim context size of 32k tokens or greater (except for Llama3), none of them can effectively handle sequence length of 32K by exceeding a qualitative threshold, Llama2-7b performance at 4K (85.6%).

Table 7: Performances (%) of ≤7B models on RULER benchmark. The performance exceeding the threshold (i.e., Llama2-7B on 4K length) is underlined.

| Model | Claimed | Effective | 4K | 8K | 16K | 32K | 64K | 128K | Avg. |
|---|---|---|---|---|---|---|---|---|---|
| Llama2-7B (Touvron et al., 2023) | 4K | | 85.6 | - | - | - | - | - | - |
| LongChat-7B (Li et al., 2023a) | 32K | <4K | 84.7 | 79.9 | 70.8 | 59.3 | 0.0 | 0.0 | 49.1 |
| Together-7B (Together.AI, 2023) | 32K | 4K | 88.2 | 81.1 | 69.4 | 63.0 | 0.0 | 0.0 | 50.3 |
| Phi3-3B (Abdin et al., 2024) | 128K | 4K | 86.7 | 78.1 | 75.6 | 70.3 | 58.9 | 43.3 | 68.8 |
| LWM-7B (Liu et al., 2024a) | 1M | <4K | 82.3 | 78.4 | 73.7 | 69.1 | 68.1 | 65.0 | 72.8 |
| ChatGLM3-6B (Du et al., 2022) | 128K | 4K | 87.8 | 83.4 | 78.6 | 69.9 | 56.0 | 42.0 | 69.6 |
| Mistral-7B (Jiang et al., 2023) | 32K | 16K | 93.6 | 91.2 | 87.2 | 75.4 | 49.0 | 13.8 | 68.4 |
| FILM-7B (ours) | 32K | 32K | 92.8 | 88.2 | 88.1 | 86.9 | 70.1 | 27.1 | 75.5 |

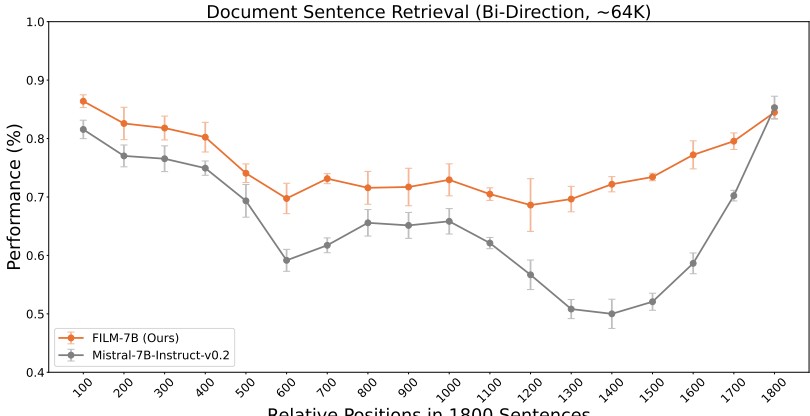

Figure 9: Performance of FILM-7B on 64K context length. The position embeddings are extended through YaRN.

Table 7 shows that FILM-7B is the first ≤7B size model that achieves 32K effective context length. Note that for the evaluation on >32K lengths (i.e., 64K and 128K), we use YaRN (Peng et al., 2023b) to extend the position embeddings without further fine-tuning.

# G   Position Extension Suffers Lost-in-the-Middle

Some existing studies focused on how to extend the context window without further training, such as YaRN (Peng et al., 2023b). We tried to extend the context window of FILM-7B from 32K to 64K with Yarn and evaluated whether the model can still overcome lost-in-the-middle problem under the extended context length. Figure 9 shows that although FILM-7B outperforms vanilla Mistral-7B on the extended 64K context window, both two models severely suffer the lost-in-the-middle problem. Such an observation indicates that simply extending the position embeddings of the model may not make the model fully utilize the extended context window.

# H Training Examples for IN2 Training

### Example 1: Fine-Grained Information Awareness (1)

**Segment:**
Last summer I took a cross-country road trip with a group of people driving different types of electric vehicles. It was an amazing experience, and our trip, our adventures, and our misadventures were portrayed in the documentary Kick Gas. That experience gave me the idea to take my own road trip to experience the freedom of traveling alone on an electric motorcycle. My goals were simple: ride the great motorcycle roads of America, meet other electric motorcycle riders, and visit friends ...

**Question:**
What were the primary goals of the individual's solo road trip on an electric motorcycle?

**Answer:**
The individual's primary goals for the solo road trip on an electric motorcycle were to ride the great motorcycle roads of America, meet other electric motorcycle riders, and visit friends.

### Example 2: Fine-Grained Information Awareness (2)

**Segment:**
Another addition to the kernel is the ar5523 driver, which was started over five years ago for the Atheros USB chipset of the same name. Extensions for supporting more chips and WLAN adapters were added to a number of other drivers; the brcmsmac WLAN driver, for example, now supports the BCM43224 Broadcom chip, while the rt2800usb RaLink driver supports the Sweex LW323 USB WLAN adapter ...

**Question:**
What new addition to the kernel supports the BCM43224 Broadcom chip?

**Answer:**
The new addition to the kernel that supports the BCM43224 Broadcom chip is the brcmsmac WLAN driver.

### Example 3: Fine-Grained Information Awareness (3)

**Segment:**
The Arlington County Board plans to vote Saturday afternoon on giving Amazon $23 million and other incentives to build a headquarters campus in Crystal City, but only after hearing scores of northern Virginia residents and advocates testify for or against the project. The five-member board is expected to support the plan, which was announced amid much hoopla on Nov. 13. The proposed county incentives are part of an agreement in which Amazon would occupy significant office space and bring at least 25,000 high-paying jobs to Arlington in coming years. Opponents hope to postpone the vote until after additional public hearings, where they want representatives of the online retail giant to answer questions directly from anyone in the community.

**Question:**
What is the Arlington County Board's plan regarding Amazon's proposed headquarters in Crystal City?

**Answer:**
The Arlington County Board is planning to vote on providing Amazon with $23 million and other incentives to construct a headquarters campus in Crystal City. However, this vote will only take place after listening to the opinions of numerous residents and advocates from northern Virginia. The board, which consists of five members, is anticipated to endorse this plan. The proposal includes Amazon occupying a significant amount of office space and introducing at least 25,000 high-paying jobs to Arlington in the forthcoming years.

## Example 4: Integration and Reasoning of Information (1)

**Segment 1:**
Cloud Cruiser's utilization analytics application enables users to manage and optimize public, private and hybrid cloud usage and spending ...

**Segment 2:**
This reduces the risk of organizations investing too much or too little in IT, eliminates unused capacity and frees up valuable IT resources for new value-adding projects ...

**Segment 3:**
As a Cloud Cruiser customer, we have seen firsthand the value that Cloud Cruiser's technology creates by enabling HPE Flexible Capacity to meter and bill for usage of on-premise IT infrastructure in a pay-as-you-go model ...

**Question:**
What benefits does the acquisition of Cloud Cruiser bring to HPE and its Flexible Capacity business?

**Answer:**
The acquisition of Cloud Cruiser enriches HPE's Flexible Capacity business by providing a utilization analytics application that allows users to manage and optimize their cloud usage and spending. Cloud Cruiser's software also facilitates the metering and billing for customers' IT consumption. This enables HPE to offer its customers an on-premise physical IT infrastructure at the cost of cloud services, effectively a rental agreement. Moreover, it reduces the risk of organizations either under or over investing in IT, eliminates unused capacity, and frees up IT resources for new projects ...

## Example 5: Integration and Reasoning of Information (2)

**Segment 1:**
Julie Annette Wasden, 45, of 38 Roles St., Haines City, was charged with retail theft at Beall's at 5998 Cypress Gardens Blvd ...

**Segment 2:**
Katherine D. Collazo Santos, 18, of 108 Hilltop Drive, Winter Haven, was charged with retail theft at Macy's at 700 Third St., S.W. ...

**Question:**
Who was charged with retail theft at Beall's and who was charged with the same crime at Macy's?

**Answer:**
Julie Annette Wasden and Katherine D. Collazo Santos

## Example 6: Integration and Reasoning of Information (3)

**Segment 1:**
A court presided by Magistrate Joseph Mifsud explained in its judgement of 17 August, 2016 in The Police v Eebis Getu, that the charges of a false criminal report and calumnious accusations may be factually similar, however, from a legal point of view they are two separate and distinct charges. The accused, Eebis Getu was charged with having filed a false police report and with not obeying a legitimate order. She admitted these charges ...

**Segment 2:**
In passing judgement the court took into consideration that the accused admitted to the charges immediately and also that she did what she did in order to be with her husband in Malta. Magistrate Mifsud referred to what Pope Francis said last June, where today̌2019s information technology brings suffering of others instantly, but we also become immune to tragedies and sufferings ...

**Question:**
What were the two charges Eebis Getu admitted to, and what was her reason for committing these actions according to the court's judgement?

**Answer:**
Filing a false police report and not obeying a legitimate order; to be with her husband in Malta.

# I Prompts For Data Generation and Training

## Example 7: Prompt For Equation 1

Generate one question and the answer from the given context. The question should be highly specific to the information provided in the context. It should not be a general question that suits any context.
Rules to follow when generate the question:
1. The question should be fully answerable from information present in given context.
2. Make sure the question is clear and unambiguous.
3. Phrases like 'based on the provided context', 'according to the context', etc, are not allowed to appear in the question.
Rules to follow when generate the answer:
1. The answer must use the information provided in the context.
2. Do not just copy words from the context. Answer the question in your own words.

\### Context \###:
$s_i$

\### Question \###:
{completion}

## Example 8: Prompt For Equation 2

Generate one question and the answer from the given context. The context contains several pieces. Answering the question should require the reader to make multiple logical connections or inferences using **at least two pieces**.
Rules to follow when generate the question:
1. The question should be fully answerable from information present in given context.
2. Make sure the question is clear and unambiguous.
3. Phrases like 'based on the provided context', 'according to the context', etc, are not allowed to appear in the question.
Rules to follow when generate the answer:
1. The answer must use the information provided in the context.
2. Do not just copy words from the context. Answer the question in your own words.

\### Context \###:
\# Piece 1: $s_i^1$
\# Piece 2: $s_i^2$
...

\### Question \###:
{completion}

## Example 9: Training Template

**Input:**
[INST] Below is a context and an instruction. Based on the information provided in the context, write a response for the instruction.

\### Context:
$\mathcal{L}_i$

\### Instruction:
$q_i$ [/INST]

**Output:**
$a_i$

# J Limitations

The main limitation of this work lies in the insufficient analysis on the choices of training hyper-parameters (e.g., learning rate, batch size, training steps and warm-up rate), data construction settings (e.g., data size and data mixture rate) and backbone models (e.g., model sizes and model architectures).

We just take the commonly used settings for IN2 training without further searching and analyzing. Intuitively, we believe the change of these settings will not affect the feasibility of IN2 training.

## K  Broader Impacts

This work used pre-trained large language models (i.e., GPT-4 and Mistral-7B) during data construction and training. Therefore, our model may inherit the potential risks of these pre-trained large language models in terms of ethical and safety issues.

