# OpenReview forum: "Make Your LLM Fully Utilize the Context"
_NeurIPS.cc/2024/Conference — NeurIPS 2024 poster_

### Official Review · Reviewer_oKze · 2024-07-12

**Soundness:** 3
**Presentation:** 3
**Contribution:** 3
**Rating:** 7
**Confidence:** 4

**Summary:**

This paper introduces INformation-INtensive (IN2) training, a data-driven approach to address the "lost-in-the-middle" problem in large language models (LLMs) with long context windows. The authors hypothesize that this issue stems from insufficient explicit supervision during long-context training. They create a synthetic long-context question-answering dataset that requires models to utilize information from various positions within long contexts. By applying IN2 training to the Mistral-7B model, they develop FILM-7B, which demonstrates significantly improved performance on probing tasks designed to assess long-context utilization across different context styles and retrieval patterns. FILM-7B also shows enhanced performance on real-world long-context tasks while maintaining comparable performance on short-context tasks. The authors provide detailed analyses of their training strategies, including the impact of sliding windows and position encoding adjustments.

**Strengths:**

- The paper introduces IN2 training, an innovative data-driven solution to address the "lost-in-the-middle" problem in long-context language models. This tackles a critical issue in the field of long-context LLMs, potentially improving their ability to utilize information from extended contexts.
- The authors develop a thorough evaluation framework called VAL Probing, which assesses long-context capabilities across various context styles (document, code, and structured-data) and retrieval patterns (forward, backward, and bi-directional). This provides a more rigorous and diverse testing suite compared to existing benchmarks like Needle-in-the-Haystack.
- FILM-7B, the model developed using IN2 training, demonstrates significant improvements not only on the probing tasks but also on real-world long-context tasks. Importantly, it maintains performance on short-context tasks, showing that the improvements in long-context handling do not come at the expense of other capabilities. The model even achieves performance comparable to or better than GPT-4-Turbo on some tasks.

**Weaknesses:**

- I do not see a major flaw in the paper. But one potential concern is: since the data is synthesized and the chunk is randomly injected into the document, would the model memorize such synthetic data but lose the generalizability on other long-context tasks beyond such scenarios?
- The authors may add more details about their construction of the training data.

**Questions:**

- Will the training data be released to the community?
- I saw a related paper a few months ago that developed a multi-hop qa style needly-in-the-haystack benchmark. I tried to find the paper during the review process but unfortunately, I didn't find it. The basic idea is similar to the training data construction method described in the paper. If the authors also came across this paper, it would be great if you could test your model on the benchmark. If not, it is totally fine. I will revise my comment and add the paper if I can find it later.

**Limitations:**

Yes.

---

> ### Author Rebuttal · Authors · 2024-08-07
>
> Thanks for your encouraging feedback on our work. We hope the following response could further address your concerns.
>
> > W1: Since the data is synthesized and the chunk is randomly injected into the Doc, would the model memorize such synthetic data but lose the generalizability on other long-context tasks beyond such scenarios?
> >
> > R1: Our evaluation results on real-world long-context tasks (Table 2) could address this concern. For instance, the long contexts in NarrativeQA are **entire books or movie scripts**, and the long contexts in Qasper are **full academic papers**. These natural long contexts are quite different from our synthetic data, and the improvements on these tasks demonstrate the generalizability beyond the synthetic pattern.
> >
> > We also further evaluate on the following few-shot learning tasks which also have different surface forms with our synthetic data. These tasks are taken from LongBench, including TREC (few-shot classification), TriviaQA (few-shot QA) and SAMSum (few-shot summarization). The improvements on these tasks further demonstrate the generalizability on general long-context tasks.
> >
> > |Model|TREC|TriviaQA|SAMSum|Avg.|
> > |---|---|---|---|---|
> > |GPT4-Turbo|77.0|91.7|39.7|69.5|
> > |Mistral|71.0|84.5|35.8|63.8|
> > |**FILM (ours)**|**76.0**|**90.0**|**39.5**|**68.5**|
>
>
> > W2: The authors may add more details about their construction of the training data.
> >
> > R2: We will add further details about the process of constructing our training data, including the selection strategy on filling segments $[r_j]$ and the data quality analysis before generating data on a large scale. Moreover, Appendix D contains details about the considerations and implementation for our segmentation process.
> > - **Selection strategy on filling segments $[r_j]$**. We randomly sample $[r_j]$ from a large candidate pool where $s_i$ and also $[s_i]$ are excluded. During sampling $[r_j]$, we restrict the distribution of len($[r_j]$) in order to manage the distribution of the final context length.
> > - **Data quality analysis**. We manually accessed 40 synthetic examples before generating data on a large scale. Some of examples are contained in our Appendix H. We find that GPT-4 is highly capable of generating QA pairs that are specifically related to the information provided in the given contexts.
>
> > Q1: Will the training data be released to the community?
> >
> > A1: Our data will be released upon completion of the internal data review process.
>
> > Q2: A related paper about multi-hop QA style Needly-in-the-Haystack benchmark.
> >
> > A2: Are you referring to the RULER benchmark [1]? This benchmark purports to access "multi-hop tracing and aggregation". Our Appendix F includes the evaluation process and results on the RULER benchmark. It shows that our FILM-7B is the first ~7B model that achieves 32K effective context length.
> >
> > |Models|Claimed Length|Effective Length|4K|8K|16K|32K|
> > |---|---|---|---|---|---|---|
> > |Llama2-7B|4K| |85.6|-|-|-|
> > |LongChat-7B|32K|<4K|84.7|79.9|70.8|59.3|
> > |Together-7B|32K|4K|$\underline{88.2}$|81.1|69.4|63.0|
> > |LWM-7B|1M|<4K|82.3|78.4|73.7|69.1|
> > |ChatGLM3-6B|128K|4K|$\underline{87.8}$|83.4|78.6|69.9|
> > |Mistral-7B|32K|16K| $\underline{93.6}$| $\underline{91.2}$|$\underline{87.2}$|75.4|
> > |**FILM-7B**|32K|**32K**|$\underline{92.8}$|$\underline{88.2}$|$\underline{88.1}$|$\underline{86.9}$|
>
> [1] RULER: What's the Real Context Size of Your Long-Context Language Models?

---

> > ### Comment · Reviewer_oKze · 2024-08-12
> >
> > Thanks for the response! Yeah, it might be the RULER benchmark. Thanks for your efforts in building this. Really hope your training data will be released to the community!

---

### Official Review · Reviewer_d47H · 2024-07-12

**Soundness:** 2
**Presentation:** 3
**Contribution:** 3
**Rating:** 6
**Confidence:** 4

**Summary:**

The paper argues that long-context models suffer from a “lost in the middle” phenomenon due to the infrequency of important data at any one position in the middle of the window during training. To correct for this, the authors propose IN2, a method of training on synthetic long-context QA data to artificially control for when important information is at each point in the context window during training. The model trained with this method, FILM-7B, demonstrates stronger performance than its base model at the 32k context scale.

**Strengths:**

S1. The paper clearly outlines a problem (the lost in the middle phenomenon), a proposed root cause (the insufficient supervision during training for important info in the middle context), and a proposed solution (IN2 training). The work is well-motivated and the proposed root cause, while not empirically demonstrated, seems reasonable.

S2. The method clearly improves the performance of the base model on both synthetic (probing) and realistic long-context tasks. It’s exciting that this improvement comes with the use of very little genuinely-long-context data, and I find it very interesting that increasing the RoPE base is helpful here.

S3. The design of the three probing tasks to focus around directionality of information is well-justified, and I believe the choice of different domains for source text for each task is appropriate, given that no general claims are being made about relative performance between “forward” and “backward” retrieval. The authors also correctly identify that these probing tasks are not sufficient for evaluation and present results on a reasonable set of existing long-context datasets.

S4. I appreciate the concern taken to avoid data contamination and the explicit description of the steps taken (in Appendix A).

**Weaknesses:**

W1. My biggest concern is that I’m not really convinced that this improvement is due to IN2 rather than the general benefits to downstream tasks of finetuning on QA and instruction tuning data, maybe coupled with some additional benefit from changing the RoPE base that would hold independently of IN2 training’s data intervention. Compared to Mistral-7b-Instruct-v0.2, FILM is trained on ~1.75M additional tokens of data, including quality instruction tuning data and a large amount of QA-style tasks. Comparing this model to Mistral-7b-Instruct-v0.2 directly does not seem to be a fair comparison for evaluating IN2’s qualities as a data selection strategy for finetuning. The real comparison here would be comparing FILM to a Mistral model finetuned on the same quantity and ordering of data but with the IN2-generated synthetic question/answer pairs removed. I understand that this is very computationally costly, but even a checkpoint at 20% of the data mixture (in the same style as the RoPE ablations) would be very helpful to understand how much benefit the method is actually providing. Ideally you could also compare finetuning on non-IN2 instruction data with and without modifying the RoPE base.

W2. The insight that increasing the RoPE base improves performance here is really cool, but I think the reasons why are underexplained and underexplored. The paper attributes this to “higher information intensity” in the context window, but this phrase is not defined in the text. I think it’s okay if the improvements from scaling theta are a purely empirical finding, but any claims about the cause of the improvement require much more justification.

W3.  Notation in 2.1. I think the level of notational formality used here for the synthetic data generation isn’t strictly necessary, but if it’s included, more specificity is necessary. In particular, is [r_j] restricted to never include s_i / [s_i]? What number of segments might be included in [s_i]? In equation 2, it may be helpful to remind the reader of what [r_i] is, as it is not referenced in the text explaining equation 2 before the equation. And when, on line 108, you write that the length balancing is achieved “by reject[ion?] sampling on [r_j]” — does this simply mean you reduce the number of segments you sample? I don’t quite understand how this is rejection sampling, especially given that all segments are the same length.

**Questions:**

Q1. There are existing QA datasets for single-hop and multi-hop reasoning. Why generate data instead of using an existing corpus and mixing in other pretraining data as the filler material?

Q2. The paper focuses on improving performance within the pretrained context length. Do you think IN2 training would be beneficial when finetuning to *extend* the context length as well?

Q3. What is information intensity (line 257), and why would training with higher information intensity require a larger RoPE base?

Q4. Can you provide any evidence from your experiments that your method's gains come from the synthetic QA data used rather than the benefits of additional (instruction tuning) data and training time?

minor suggestions/typos:

- line 205: `lm_eval` provides a preferred citation format; please cite them instead of using a footnote here.
- line 554: “illustrates how do we segment” → “illustrates how we segment”
- line 556: capitalized “We” in the middle of the sentence
- line 127: was it 300 GPU days for all experiments or for training a single model? It seems from Table 3 that there were multiple models (at least partially) trained; it would be helpful to have numbers for total compute used and compute used on the final FILM-7b.
- Needle in the Haystack is often attributed jointly to [Ivgi et al 2023](https://aclanthology.org/2023.tacl-1.17/) and [Liu et al 2023](https://arxiv.org/abs/2307.03172).
- Table 1: why are the FiLM numbers for database bolded when GPT-4-Turbo is outperforming it?
- It would be nice to reference what kinds of experiments are in the appendices somewhere in the main text, to make them more discoverable (e.g. I don’t think Appendix G is mentioned anywhere in the main body).
- In the second critiquing needle in the haystack, should cite work on the reversal curse and other critiques of needle in the haystack (there have been several, including critique specifically of the single-needle case). Some of this work is already discussed in the related work, but should be cited here to show that there is community discussion about these critiques of needle in the haystack.
- The claim that sliding window strategies greatly hurt long-context capability— I think this is a bit too broadly stated for the experimental results provided.
- Line 70: “This demonstrates that training on synthesized long-context data can be generalized to real-world scenarios.” I think care should be taken here to emphasize that this is *additional* training on synthesized long-context data, after already training on real long-context documents at that same length (during Mistral’s pretraining).

**Limitations:**

The authors identify insufficient ablations as a limitation, which I agree with. However, the concern of the limitations section seems to be that the method might either 1) be brittle to the choice of hyperparameters here (which I agree is unlikely) or 2) perform much better with careful tuning. My concern is really 3) the method may not outperform existing baselines of normal instruction tuning (possibly plus RoPE base scaling).

---

> ### Author Rebuttal · Authors · 2024-08-07
>
> Thanks for your detailed review and constructive suggestions. We hope the following response and our additional experiments have addressed your concerns.
>
> > W1 and Q4: Require experimental comparison with normal instruction tuning.
> >
> > R1: We present further experimental results to demonstrate that **normal instruction tuning (IT) cannot alleviate the lost-in-the-middle problem**. We conduct a comparison between the effectiveness of IN2 training and normal instruction tuning, with the results shown in the following table. Specifically, **both Mistral+IN2 and Mistral+IT have the same training data size** (20% of our full data size, ~300K examples) and do not incorporate adjustment of RoPE. The training data for Mistral+IT are sampled from a large-scale instruction-following dataset OpenOrca [1]. Compared with Mistral+IN2, the gains from normal instruction tuning are marginal and unstable.
> >
> > |  | Doc| | Code| | DB| | All| |
> > |---|---|---|---|---|---|---|---|---|
> > |Model|Avg|Gap&darr;| Avg|Gap&darr;| Avg|Gap&darr;| Avg|Gap&darr;|
> > |Mistral|74.2|32.1|20.3|59.5|47.5|77.0|47.3|56.2|
> > |Mistral+IT|69.0|25.9|30.2|76.5|53.4|54.4|50.9|52.3|
> > |**Mistral+IN2 (ours)**|**82.9**|**11.5**|**74.5**|**27.7**|**83.5**|**31.6**|**80.3**|**23.6**|
> >
> > Furthermore, the PDF file in our general response illustrates the performance curves of Mistral+IT. These comparisons demonstrate that IN2 training can effectively alleviate the lost-in-the-middle problem while the normal instruction tuning cannot.
>
> > W2 and Q3: Require further explanations for “higher information intensity” and the insights behind adjusting RoPE.
> >
> > R2: The term "higher information intensity" is not a quantitative measure but a conceptual phrase we utilize to characterize our IN2 training. In our QA-style training data, **the essential information for different questions are spread evenly across all positions, instead of being concentrated only at the beginning and the end (which would indicate high sparsity)**.
> >
> > The rationale behind our adjustment of RoPE is derived from practical experiences with context-extension training. Studies [2,3] show that when the context window was extended from 4K to 32K during training, the RoPE base $\theta$ was increased from 10,000/50,000 to 1,000,000. There is a high-level similarity between context-extension training and our IN2 training: **both aim to enhance the model’s capability to perceive information from a wider range of input positions** compared to the previous training stages. Consequently, we carry out experiments to investigate whether the experience from context-extension training could also benefit our IN2 training.
>
>
>
> > W3: Require further clarification on notations and sampling process.
> >
> > R3: We will revise our Section 2 to make it clearer for readers.
> > - About $[r_j]$ and $s_i$/$[s_i]$: It is satisfied that $[r_j]$ will not contain $s_i$/$[s_i]$. Moreover, we make a restriction that 2 <= len($[s_i]$) <= 16. We set an upper bound here considering the time cost of calling GPT-4 for an excessively long input sequence.
> > - About the sampling process for length balance: There is a typo in line 108: the “reject sampling on $[r_j]$” should be “restricted sampling on $[r_j]$”. During sampling $[r_j]$, we restrict the distribution of len($[r_j]$) in order to manage the distribution of the final context length. Thanks for pointing out this error!
>
> > Q1: Why synthesize QA data instead of using an existing QA dataset?
> >
> > A1: There are mainly two reasons.
> > - First, the use of synthetic data allows our methodology to be **scalable in terms of accessing larger training data sizes**. Moreover, it offers **flexibility in training a long-context model in a specific domain** where there may be a scarcity of manually annotated QA pairs. Leveraging synthetic training data has recently become a prevalent data construction strategy for training LLMs [4,5].
> > - Second, as we have taken many existing QA datasets in our evaluations (including both long-context and short-context datasets), by utilizing synthetic data exclusively, we can **circumvent potential data contamination** issues between training and evaluation.
>
> > Q2: Would IN2 training be beneficial to extend the context length?
> >
> > A2: That's an insightful question. We think the answer is Yes, and we have some preliminary results and observations: applying IN2 training on a 32K context window and subsequently extending it to 64K with YaRN (without fine-tuning) still yields improvements (as illustrated in Figure 8 of our Appendix). These results may not be as robust as within the 32K context, but we believe that incorporating fine-tuning on the 64K context window could further enhance the performance.
>
> [1] https://huggingface.co/datasets/polinaeterna/OpenOrca.
>
> [2] InternLM2 Technical Report
>
> [3] Qwen2 Technical Report
>
> [4] Phi-3 Technical Report: A Highly Capable Language Model Locally on Your Phone
>
> [5] The Llama 3 Herd of Models

---

> > ### Comment · Reviewer_d47H · 2024-08-12
> >
> > Thanks for the additional results and the response!
> >
> > >  demonstrate that normal instruction tuning (IT) cannot alleviate the lost-in-the-middle problem
> >
> > Thanks for these results! This addresses my primary concern.
> >
> > > "higher information intensity" is not a quantitative measure but a conceptual phrase
> >
> > I think if you're going to use this term, you should define it with some degree of rigor. It sounds like this concept could be described also as "more uniform information density," which is a more commonly used term.
> >
> > > rationale behind our adjustment of RoPE
> >
> > This intuition here is interesting, and would be a good addition to the paper!
> >
> > > as we have taken many existing QA datasets in our evaluations (including both long-context and short-context datasets), by utilizing synthetic data exclusively, we can circumvent potential data contamination issues
> >
> > It's worth noting that this doesn't completely avoid issues of data contamination-- synthetic data generated from GPT4 could leak some information from datasets that GPT4 saw during pretraining-- but data contamination from pretraining is complex and this is a good-faith attempt to avoid it.
> >
> >
> > While I think the paper could still benefit from some revisions before camera-ready, my primary concern about the method is addressed and I think the work is sound, so I'll raise my score 4 -> 6.

---

### Official Review · Reviewer_UZJA · 2024-07-12

**Soundness:** 2
**Presentation:** 3
**Contribution:** 3
**Rating:** 6
**Confidence:** 4

**Summary:**

In this paper, the authors propose two types of training to mitigate the lose-in-the-middle issue of Large Language Models (LLMs). The first one creates some QA pairs. They first pick 128-token segments with which they use LLM to generate QA pairs and mix these 128 tokens with other contexts to form long context QA pairs. The second one extends this approach with multiple segments. The authors also propose three VAL Probing tasks that ask LLMs to print the context of certain phrases. Results show that the model fine-tuned with proposed methods can significantly mitigate the lose-in-the-middle issue on challenging long context tasks and do well on Val Probing while keeping good performance on short context tasks.

**Strengths:**

1. Lost-in-the-middle is an important and challenging tasks that need to be solved to fully utilize the capability of LLMs.
2. The proposed fine-tuning approaches demonstrate effective performance on many real-world problems, showing the potential usage of the proposed method on Large Language Models (Perhaps GPT-4). Also, it can also keep the good performance on short tasks, showing the fine-tuning won't hurt the unrelated tasks
3. The paper is well-organized and easy to follow. I like figure one which shows the core idea of whole paper.

**Weaknesses:**

1. The comparison with natural dataset. What is the difference between fine-tuning on existing long context tasks, such as L-EVAL, LongBench, and SCOLLS. Why do you use synthetic data rather than these data? Do you think your synthetic data can work better? This comparison should be helpful in understanding the effectiveness of proposed approaches.

2. Many choices are not justified.  What is the context you use to mix 128-token segments with gold answers? Are they the best choices compared with using similar contexts like LongBench does? Do you have any criteria for choosing segments with gold answers?

3. Intermediate results are not evaluated. What is the quality of the synthetic QA?  Can we ensure the multihop question is valid?  Is the split of two or more segments natural (not breaking the context)? Which fine-tuning approach helps more on the final results? Some quality evaluation and ablation should be important for the analysis of the proposed approaches.

4. Lacks detail analysis. For instance, the fine-tuning is only conducted on QA tasks. How could it improve the performance of Summarization? Will you include more content in the middle of the source as well?

Overall, the strong improvement in small LLM demonstrates the effectiveness of the proposed method. Thus, I would assume that the proposed approaches can obtain positive answers to most of the questions above.

**Questions:**

See weakness.

Also, what is bold in table 2?

**Limitations:**

They have limitations and broader impact sections. I think more discussion on synthetic data usage will be helpful.

---

> ### Author Rebuttal · Authors · 2024-08-07
>
> Thanks for your insightful questions about our work. The following are our responses to these questions.
>
> > Q1: What are the reasons for using synthetic data rather than existing natural long context datasets during fine-tuning?
> >
> > A1: There are mainly two reasons.
> > - First, many existing natural long context datasets are often purposed for evaluation rather than training, including datasets such as L-EVAL [1], LongBench [2], and SCOLLS [3]. Therefore, by utilizing synthetic data exclusively, we can **circumvent potential data contamination** issues between training and evaluation.
> > - Second, the use of synthetic data allows our methodology to be **scalable in terms of accessing larger training data sizes**. Moreover, it offers **flexibility in training a long-context model in a specific domain** where there may be a scarcity of manually annotated QA pairs. Leveraging synthetic training data has recently become a prevalent data construction strategy for training LLMs [4,5].
>
> > Q2: What is the context we use to mix 128-token segments with the gold answer?
> >
> > A2: When selecting the filler segments $[r_j]$ to create a lengthy context, we want that $[r_j]$ bears no connection to the QA pair. This way, to respond to the given question, **the model will be trained to carefully find out and utilize the related $s_i$ (or $[s_i]$) from a large amount of redundant information**. For implementation, we randomly sample $[r_j]$ from a large candidate pool where $s_i$ (or $[s_i]$) are excluded. We have manually inspected 40 synthetic cases to verify that this random selection generally meets our expectations of unrelatedness.
> >
> > We did not use the similarity-based selection strategy due to the following concerns:
> > - Similar segments could have supporting information for the QA pairs, **creating a shortcut during training**. Given that $[r_j]$ could also provide useful information to answer the question, the model may just use the $[r_j]$ at the beginning and end of the context, without thoroughly searching for the $s_i$ in the middle.
> > - Similar segments might contain information that could lead to **ambiguity in answering the question** (for instance, different individuals with the same name). This ambiguity could potentially affect our training.
> >
> > It is a valuable research question to explore how to properly incorporate the similarity-based selection with our IN2 training. Mitigating the above issues requires further designs and ablations, such as carefully controlling the mixture proportions, designing and adjusting the rule-based/embedding-based filters, and also incorporating human check if necessary. Given the high training cost, we consider the use and ablation of similarity-based selection for future work.
>
> > Q3: Require further analysis on data quality and fine-tuning approach.
> >
> > A3:
> > - For the data quality, **a manual assessment of 40 synthetic examples was conducted prior to large-scale data generation** (20 for local information QA and 20 for multi-hop QA). Some of these examples are included in Appendix H. We find that GPT-4 is highly capable of generating QA pairs that are specifically related to the information provided in the given contexts. To generate multi-hop QA pairs, a simple prompting strategy was employed that utilizes ** to emphasize the need for "using at least two segments" to generate the QA pair (refer to Appendix I). Using this prompt, all 20 sampled multi-hop QA pairs have successfully incorporated information from at least two segments.
> >
> > - For the fine-tuning approach, we take the **standard instruction-tuning approach** for fine-tuning, which involves updating the model parameters only based on the loss on the output side. We do not include the input-side loss on the synthesized long contexts because we are concerned that it will dominate the training process and interfere with natural language modeling. For hyper-parameters, we initially tried four learning rates (1e-5, 5e-6, 1e-6 and 5e-7) in the first 100 training steps. We chose the 1e-6 learning rate as it exhibited the best loss trend in the first 100 training steps. Other hyper-parameters in Section 2.2 follow the general settings for post-training.
>
> > Q4: Why fine-tuning on QA-style data can enhance performance in other long-context tasks such as summarization?
> >
> > A4: Although the surface forms differ across various tasks, we believe that **the underlying capability to comprehend long-context information is transferable**. Our further evaluation results on few-shot learning tasks can also demonstrate this transferability. These tasks include TREC (few-shot classification), TriviaQA (few-shot QA) and SAMSum (few-shot summarization). Although the surface form of few-shot learning is quite different from our zero-shot training data, these results further support that an improved capability to perceive long-context information can be generalized beyond surface forms.
> >
> > |Model|TREC|TriviaQA|SAMSum|Avg.|
> > |---|---|---|---|---|
> > |GPT4-Turbo|77.0|91.7|39.7|69.5|
> > |Mistral|71.0|84.5|35.8|63.8|
> > |**FILM (ours)**|**76.0**|**90.0**|**39.5**|**68.5**|
>
> > Q5: What are bold in Table 2?
> >
> > A5: The bold numbers are SOTA-level results among ~7B open-source models. Specifically, for each task, we bold the highest result and the results within a margin to the highest one (0.5 for summarization tasks and 2.0 for others).
>
> [1] L-Eval: Instituting Standardized Evaluation for Long Context Language Models
>
> [2] LongBench: A Bilingual, Multitask Benchmark for Long Context Understanding
>
> [3] SCROLLS: Standardized CompaRison Over Long Language Sequences
>
> [4] Phi-3 Technical Report: A Highly Capable Language Model Locally on Your Phone
>
> [5] The Llama 3 Herd of Models

---

> > ### Comment · Reviewer_UZJA · 2024-08-12
> >
> > Thanks for the in-depth analysis and reply the concerns one by one! I hope the authors can also include the results and analysis in the final version. Thanks!

---

### Official Review · Reviewer_HMgi · 2024-07-13

**Soundness:** 3
**Presentation:** 3
**Contribution:** 3
**Rating:** 8
**Confidence:** 3

**Summary:**

This work tackles an essential problem in LLMs, called lost-in-the-middle, that causes the lack of information in the middle of lengthy input. As a data-driven solution, the authors propose a new training approach, INformation-INtensive (IN2). IN2 aims to enhance the use of context information in an unbiased position by shuffling the clue part of the data during training. When creating the shuffled data, the authors use GPT-4-Turbo to maintain its grammatical and semantic appropriateness. Furthermore, the authors propose a new model, FILM-7B, additionally trained Mistral-7B on a dataset created by their proposed IN2. To comprehensively evaluate FILM-7B's long-context handling ability, the author introduces VArious Long-context (VAL) Probing. VAL Probing results show that FILM-7B can outperform even GPT-4-Turbo by avoiding being trapped by lost-in-the-middle. Experimental results on long-context tasks from the LongBench collection and short-context tasks, MMLU, BoolQ, RACE-High, CommonsenseQA, ARC-Challenge, HellaSwag, GSM8K, and MATH show that FILM-7B outperforms Mistral-7B models.

**Strengths:**

- This paper proposes a new training method, INformation-INtensive (IN2), that can make trained LLMs handle long-context with mitigating the influence of lost-in-the-middle.
- By using IN2, the authors train Mistral-7B to make a new model, FILM-7B, than can handle long-context.
- The authors provide a newly constructed evaluation task, VArious Long-context (VAL) Probing, that covers various long-context tasks.
- The evaluation result on VAL Probing shows the effectiveness of IN2 based on the improved performance of FILM-7B.
- The evaluation results on long-context and short-context tasks also show that FILM-7B outperforms FILM-7B.

**Weaknesses:**

- From the viewpoint of Scaling Laws, comparing Mistral-7B and its further trained model FILM-7B is unfair.

**Questions:**

Currently, the evaluation is conducted in a zero-shot setting. Is there any possibility that FILM-7B outperforms proprietary models like GPT-4-Turbo on a few-shot setting by using its ability to handle long contexts equally?

**Limitations:**

Even though IN2 mitigates the lost-in-the-middle problem, experimental results show that this problem is still left in some tasks.

---

> ### Author Rebuttal · Authors · 2024-08-07
>
> Thanks for your valuable feedback on our work. We hope the following response could further address your concerns.
>
> > W1: From the viewpoint of Scaling Laws, comparing Mistral-7B and its further trained model FILM-7B is unfair.
> >
> > R1: We present further experimental results to compare IN2 training and normal instruction tuning (IT) **under the same training data size**. These results demonstrate that **IN2 training can effectively alleviate the lost-in-the-middle problem while the normal instruction tuning cannot**. In the following table, both Mistral+IN2 and Mistral+IT have the same training data size (20% of our full data size, ~300K examples). Furthermore, the PDF file in our general response illustrates the performance curves of Mistral+IT. Compared with Mistral+IN2, the gains from normal instruction tuning are marginal and unstable.
> >
> >   |  | Doc| | Code| | DB| | All| |
> >   |---|---|---|---|---|---|---|---|---|
> >   |Model|Avg|Gap&darr;| Avg|Gap&darr;| Avg|Gap&darr;| Avg|Gap&darr;|
> >   |Mistral|74.2|32.1|20.3|59.5|47.5|77.0|47.3|56.2|
> >   |Mistral+IT|69.0|25.9|30.2|76.5|53.4|54.4|50.9|52.3|
> >   |**Mistral+IN2 (ours)**|**82.9**|**11.5**|**74.5**|**27.7**|**83.5**|**31.6**|**80.3**|**23.6**|
>
>
> > Q1: How well does FILM-7B perform on few-shot setting?
> >
> > A1: We conduct further evaluations on the few-shot learning tasks in LongBench, including TREC (few-shot classification), TriviaQA (few-shot QA) and SAMSum (few-shot summarization). Although FILM-7B does not outperform GPT-4-Turbo, it shows that **IN2 training can significantly close the gap with proprietary models on few-shot settings**. These results demonstrate that even though the surface forms differ across various long-context scenarios, we believe that **the underlying capability to comprehend long-context information is transferable**.
> >
> > |Model|TREC|TriviaQA|SAMSum|Avg.|
> > |---|---|---|---|---|
> > |GPT4-Turbo|77.0|91.7|39.7|69.5|
> > |Mistral|71.0|84.5|35.8|63.8|
> > |**FILM (ours)**|**76.0**|**90.0**|**39.5**|**68.5**|

---

> > ### Comment · Reviewer_HMgi · 2024-08-12
> > **The response meets my expectations**
> >
> > I appreciate your in-depth responses based on the actual evaluation results. The result addressed every concern raised by me. I am really thankful to the authors for doing such high-quality research.

---

### Author Rebuttal · Authors · 2024-08-07

We thank all reviewers for their constructive comments and questions on our paper.
We appreciate the recognition of the important and challenging research question we explored, the innovative idea and the effectiveness method we proposed, and the thorough evaluations we conducted.

We summarize our responses to the main concerns as follows and will include all other minor points in our revised version.

- **More experimental results and comparisons.** We present additional results to further demonstrate the effectiveness of IN2 training.
    - Comparison with normal instruction tuning under the same data size (R1 to Reviewer HMgi, R1 to Reviewer d47H and Figure 1 in PDF file).
    - Performance of FILM-7B on few-shot scenarios (A1 to Reviewer HMgi, A4 to Reviewer UZJA and R1 to Reviewer oKze).
    - Performance of FILM-7B on RULER benchmark (A2 to Reviewer oKze).

- **Why do we use synthetic data?** There are mainly two reasons.
    - First, the use of synthetic data allows our methodology to be **scalable in terms of accessing larger training data sizes**. Moreover, it offers **flexibility in training a long-context model in a specific domain** where there may be a scarcity of manually annotated QA pairs. Leveraging synthetic training data has recently become a prevalent data construction strategy for training LLMs (e.g., Phi-3 and Llama-3.1).
    - Second, exclusively utilizing synthetic data for training can **circumvent potential data contamination** issues in our evaluations on natural language datasets, including both long-context and short-context datasets.

- **More details about data construction and data quality analysis.** We will add further details about the process of constructing our training data, including the selection strategy on filling segments $[r_j]$ (A2 to Reviewer UZJA, R3 to Reviewer d47H and R2 to Reviewer oKze) and the data quality analysis before generating data on a large scale (A3 to Reviewer UZJA and R2 to Reviewer oKze). Moreover, Appendix D contains details about the considerations and implementation for our segmentation process.

---

### Decision · Program_Chairs · 2024-09-25

**Decision:**

Accept (poster)

**Comment:**

The paper identifies an issue with LLMs, commonly referred to as lost-in-the-middle problem, where they exhibit positional bias when integrating information from in-context document, often ignoring information presented in the middle. The core idea is to generate large-scale synthetic question answering dataset (1.1M long-context data) where important information is evenly spread throughout the context to continue training LLMs. In addition, they also propose improved evaluation setting (forward, backward, bi-directional information retrieval) and evaluate their approach extensively. The proposed method shows performance gains on a variety benchmark, both long-context focused and regular text understanding/reasoning benchmark.

The reviewers have pointed out some weaknesses with the initial draft, such as (1) unfair with base model which were trained on fewer tokens (reviewer HMgi, d47H) and (2) the assessment of generated QA data quality (Reviewer UZJA). The authors have provided additional experimental results and addressed most, if not all, concerns of the reviewers.

For camera-ready, if time/compute allows, it would be great to have an experiment showing results on performing continued fine-tuning on another base model to show the generality of the proposed method. Overall the paper presents a convincing approach to address well-known issue in LLM.